# Quantitative proteomic analysis identified differentially expressed proteins with tail/rump fat deposition in Chinese thin- and fat-tailed lambs

Jilong Han[1,2,3], Tingting Guo[1,4], Yaojing Yue[1,4], Zengkui Lu[1,4], Jianbin Liu[1,4], Chao Yuan[1,4], Chune Niu[1,4], Min Yang[2,3]*, Bohui Yang[1,3]*

**1** Lanzhou Institute of Husbandry and Pharmaceutical Sciences, Chinese Academy of Agricultural Sciences (CAAS), Lanzhou, China, **2** College of Animal Science and Technology, Shihezi University, Shihezi, China, **3** Key Laboratory of Animal (Poultry) Genetics Breeding and Reproduction, Ministry of Agriculture and Rural Affairs, Institute of Animal Science, CAAS, Beijing, China, **4** Engineering Research Center of Sheep and Goat Breeding, CAAS, Lanzhou, China

* yangmin2017@shzu.edu.cn (MY); yangbohui@caas.cn (BY)

**Data Availability Statement:** All relevant data are within the paper and its Supporting Information files.

## Abstract

Tail adipose as one of the important functional tissues can enhance hazardous environments tolerance for sheep. The objective of this study was to gain insight into the underlying development mechanisms of this trait. A quantitative analysis of protein abundance in ovine tail/rump adipose tissue was performed between Chinese local fat- (Kazakh, Hu and Lanzhou) and thin-tailed (Alpine Merino, Tibetan) sheep in the present study by using lable-free approach. Results showed that 3400 proteins were identified in the five breeds, and 804 were differentially expressed proteins, including 638 up regulated proteins and 83 down regulated proteins in the tail adipose tissues between fat- and thin-tailed sheep, and 8 clusters were distinguished for all the DEPs' expression patterns. The differentially expressed proteins are mainly associated with metabolism pathways and peroxisome proliferator activated receptor signaling pathway. Furthermore, the proteomics results were validated by quantitative real-time PCR and Western Blot. Our research has also suggested that the up-regulated proteins ACSL1, HSD17β4, FABP4 in the tail adipose tissue might contribute to tail fat deposition by facilitating the proliferation of adipocytes and fat accumulation in tail/rump of sheep. Particularly, FABP4 highly expressed in the fat-tail will play an important role for tail fat deposition. Our study might provide a novel view to understanding fat accumulation in special parts of the body in sheep and other animals.

## Introduction

Sheep (*Ovis aries*) domesticated during the Neolithic revolution approximately 9000 years ago is an important livestock species for meat and agricultural products worldwide [1]. Since domestication, sheep have established in a wide geographical range due to their adaptability to

**Funding:** This research was funded by the National Natural Science Foundation of China, grant numbers 31702092, 31802045 and 32060745; Modern China Wool & Cashmere Technology Research System, grant number CARS39-02; Agricultural Science and Technology Innovation Program for Genetic Resource and Breeding of Fine-Wool Sheep, CAAS, grant number ASTIP-2015-LIHPS; Special Funds for Basic Research of Young Scholars of Shihezi University, grant numbers CXBJ201903, CXRC201808. All these founding directly or indirectly provide the transportation fee for sample collection, labor costs, kit and reagents costs, and mass spectrometry analysis costs, as well as the publication fee of this research.

**Competing interests:** The authors have declared that no competing interests exist.

poor nutrition diets, tolerance to extreme climatic conditions. The tail fat deposition is one of the obvious morphological changes in sheep, which is considered to be an adaptive response to the hazardous environment, thus fat-tail is a valuable energy reserve for the animal during migration and drought food deprivation [2]. Tail adipose is one of the important functional tissues for sheep, and it is a special organ or part for energy storage, similar to a camel's hump [3], and it has an unique tolerance for extreme drought and desert conditions. However, the wild ancestor of sheep was thin tail, and the fat-tail phenotype was developed between 3000 and 1500 BC in the Fertile Crescent [4]. Sheep with adipose deposits in the tail were particularly selected for breeding purposes in the history [5]. Fat-tailed and fat-rumped sheep are characteristic of semi-arid environments and commonly found worldwide, including in Eastern and Southern Africa, the steppes of Central Asia, as well as numerous countries in the Middle-East [6]. Nowadays, fat-tailed sheep (FTS) comprise approximately 25% of the world's sheep breeds [4]. In China, fat-tailed breeds comprise 80% of the total population and most of these native domestic sheep breeds are highly adapted to local environmental conditions [7]. The high amount of fat accumulation in the tail can play an important role as a survival mechanism for FTS during periods of food shortage and as a valuable reserve [8, 9]. However, in recent years, most of the advantages of large fat-tail have reduced, the large fat tails have lost their importance in recent decades, which also can reduce meat value, producers and consumers are interested in lower-fat meat [3, 10]. Therefore breeding for reducing tail fat deposition is becoming an important goal toward better profitability in the sheep industry [11].

Genetic effects influencing fat deposition in different body sites are still an important area of investigation to select desirable livestock for breeding and production [12, 13]. As we know, this trait is one of the most challenging areas of research in most countries grazing fat tail sheep breeds. We believe that the fat tissue in fat-tail or fat-rump is a good model to study the mechanism of fat deposition in animal. To date, several investigations into the inheritance of tail types have been undertaken to determine the genetic variation associated with this phenotype on various sheep populations by comparing allele frequencies of fat-tail with thin-tail breeds, and some candidate genes were proved to be associated with the tail phenotype of sheep, including *PPP2CA*, *EBP*, *PPP1CC*, *PDGFD*, *BMP2*, *VGR* and *VNRT* [7, 13–24], among them, *PDGFD* has the strongest selection signal in different tailed sheep worldwide, indicating *PDGFD* may be involved in fat deposition in sheep tail [23, 24]. Nevertheless, the molecular mechanism underlying fat-tail development remains to be elucidated in sheep. On the other hand, recent studies have focused on gene expression profile in these special sheep adipose tissue, and the HOX-related genes, *FABP4*, *FABP5*, *SCD*, *CREB1*, *WDR92* and *ETAA1* were identified to be potentially associated with the fat deposition in sheep tails [3, 25–30]. Recently, miRNAs also have been found with potential roles in regulating adipose metabolism, some miRNA and their target genes play a key role in fat deposition in sheep [29, 31, 32]. Recent studies have shown that non-coding RNAs (ncRNAs) might be driving fat deposition [2, 33]. These studies laid the foundation for studying the formation of sheep tail fat phenotype. However, up to now, the regulatory mechanism of fat-tail development have not been well characterized, especially there were no study using the tail fatty tissues of lambs, and the early growth stage could be an important period for tail fat deposition. Therefore, we believe that the potential underlying molecular mechanism for tail or rump fat accumulation is very complex. It would be necessary to use new method to identify candidate genes and molecular pathways regulating tail fat deposition in sheep.

Comparative proteomics are an efficient and accurate choice for measuring protein expression that has been widely employed for large-scale studies on complex traits in many livestock. As the protein equivalent of genomics, it provides the foundation for constructing and extracting useful knowledge on phenotypic traits research [34]. To the best of our knowledge, few

studies on comparative study of adipose between FTS and TTS worldwide by proteomics-based approach. Toward better understanding of the molecular mechanisms that might regulate fat-tail development. In this study, five Chinese typical sheep breeds with different tail type were selected, and we conducted a comparative proteomic approach in attempt to elucidate the proteins and molecular pathways involved in fat-tail development. In addition, our study might provide a novel view to understand the molecular mechanism of fat-tail development or fat deposition in other animals.

## Materials and methods

### Animals, adipose tissues collection and ethics

Five Chinese indigenous local sheep breeds with different tail type were selected, including three FTS breeds, Kazakh sheep (fat-rumped) obtained from Hami, located in the southeast of Xinjiang province; Lanzhou big FTS (with long fat tail) raised in the countryside of Lanzhou city, Gansu province; Hu sheep (short fat-tailed) come from Yongchang city, Gansu province, and two TTS breeds, Alpine Merino (long thin-tailed) and Tibetan sheep (short thin-tailed) provided by Huangcheng Breeding Farm, located in the northwestern Gansu province close to the Qilian mountain ranges (Fig 1A). Three male lambs from each purebred breeds were selected at birth and all the lambs were unrelated individuals. Lambs were located indoors with their dams and all of them were not weaning, reared under similar environmental conditions with supplemental cracked corn and dry alfalfa and allowed free access to feed and water under natural lighting. The required tail adipose tissue (TAT) samples were collected immediately from the same position in the middle of the tail/rump after the lambs were sacrificed (distributed at periods of April 20, 2014-April 28, 2014; 13–15 kg live weight; 40±10 days on average), which were quickly placed into liquid nitrogen for further analysis.

All animal experiments used in this study were carried out in strict accordance with the animal ethics procedures and guidelines of institutional animal care and use committee of Lanzhou institute of husbandry and pharmaceutical sciences (CAAS), and we were also granted permission to complete this study and the legal certificate number was SCXK (Gan 2014–0002).

### Adipose tissue staining

The adipose tissue staining was performed by the method described by Ruoss [35]. In brief, TATs were cut into small sections, then they were embedded in paraffin using OTC compound (SAKURA Tissue-Tek, USA) after fixation at -80°C for 12 h. Adipose tissues were sliced with the freezing microtome (Leica CM 1860UV, Germany) and serially sectioned (12 μm). The adipocyte size was observed under a light microscope (Olympus BX43 and DP26, Japan) after oil red O staining. The diameter of adipose cells was measured using the cellSens entry image analysis system.

### Protein extraction of adipose tissues and peptide preparation

Proteins were extracted from samples after the fatty acids were isolated with chloroform and methanol (1:2) according to the previous method [36], and then treated according to the description in Fig 1B. Briefly, tissues were ground into a fine powder in liquid nitrogen, and 200 mg from each replicate samples was extracted with the lysis buffer (8 M urea, 2 M CHAPS, 4% CHAPS, 20 mM Tris-base and 30 mM DTT). The protein extract was centrifuged at 15,000 g at 4°C for 20 min. The above collected supernatant was transferred to a new tube, and proteins were precipitated in ice-cold acetone with three volumes at -4°C for 1.5 h and centrifuged at 15,000 g for 10 min at 4°C. The supernatant was discarded, and 40 mM $NH_4HCO_3$ solution

**Fig 1. Proteome analysis of isolated TAT in five sheep breeds.** (A) The imagine of the selected 5 lambs, Hu sheep, Lanzhou big fat tailed sheep, Kazakh sheep, Alpine Merino and Tibetan sheep, respectively. (B) The flowchart shows the experimental design for comparative proteomics of TAT in five local breeds. (C) The Venn diagram shows the overlap of the identified proteins of five breeds.

was added to the pellets. After the protein concentration was quantified, the flow-through was discarded from the collection tube. Then proteins were reduced with 5 mM DTT at room temperature for 30 min, cooled to room temperature, alkylated with 20 mM IAA in a darkroom for 30 min, and 5 mM DTT was added and incubated in a darkroom for 15 min. The reduced

and alkylated proteins were digested using trypsin (Promega, USA) in a volume ratio of 1:50 (enzyme/protein) at 37°C for 12–14 h. The enzymatic digestion was stopped by adding 1 μl of formic acid, and the samples were then vacuum-dried using a SpeedVac system (RVC 2–18, Marin Christ, Osterod, Germany).

## LC–MS/MS analysis

Ultimate nano HPLC 1000 system coupled with QExactive quadrupole-orbitrap mass spectrometer (Thermo Fisher Scientific, Germany) was utilized for label-free analysis according to the previously described method [37]. Digested peptides with 500 ng were loaded on a 2 cm long, 100 μm inner diameter fused silica trap column containing 5.0 μm aqua C18 beads (Thermo Fisher Scientific) at a flow rate of 4 μl/min prior to analytical separation. Then, peptides were separated on a column packed with 2 μm C18 (100 Å, 75 μm×50 cm, Thermo Fisher Scientific) at a flow rate of 350 nl/min, in which the eluents were solvent A (contained 0.1% (v/v) formic acid (FA) in $H_2O$) and solvent B (contained 0.08% (v/v) FA in acetonitrile/$H_2O$ (80% 20% : v/v)). Chromatographic conditions of gradient elution went from 3% B up to 8% B in 5 min, from 8% B up to 20% B in 80 min, from 20% B up to 30% B in 20 min, then from 30 B up to 90% B in 5 min, followed by an increase to 100% B for an additional 10 min. The total procedure time was 120 min and thrice for each sample.

MS survey scan was obtained for the m/z (mass-to-charge ratios) range 350–1,800 with a resolution of 35,000 at m/z 400. Ion signals were collected in a data-dependent mode with the following settings: full scan resolution at 70,000 MS/MS scan resolution at 17,500 isolation window: 2 m/z. The MS/MS data were acquired in raw files using the Xcalibur software (version 2.2, Thermo Fisher Scientific).

## Quantitative proteomic analysis

The generated raw data from label-free LC-MS/MS were further examined by PEAKS software (version 7.5, Bioinformatics Solutions, Waterloo, Canada) as our previously described method [38]. Database searches were performed against the Databases (containing 22,822 protein sequences of *Ovis aries*, downloaded from Ensemble) with a peptide mass tolerance of ±15 ppm and fragment ion tolerance of 0.05 Da. The proteins were cleaved by trypsin, and the two missed cleavages were accepted. Label-free approach was performed to detect the relative quantification by Q module in PEAKS. Peptide features and proteins were considered to be significant between different samples when $p$ value of <0.05 and a fold change of ≥2. To identify the similar expression pattern of DEPs in FTS vs. TTS, the log10 (Peaks area + 1) fold changes for all 804 DEPs were subjected to hierarchical analysis using the 'hclust' function in R (version 3.2.5, http://cran.r-project.org/). The cluster dendrogram was divided using the 'complete' function to classify the proteins for characterization based on the change in expression.

## Bioinformatics analysis

Gene Ontology (GO) enrichment analysis of DEPs was implemented using g:Profiler (http://biit.cs.ut.ee/gprofiler/) as described previously [39], and all the 3,400 proteins detected in the five samples were chosen as the background proteins. Protein classification was performed based on their functional annotations. In addition, to identify the biological pathways that were involved in fat deposition, the statistically significant for GO and Kyoto Encyclopedia of Genes and Genomes (KEGG) biological pathways of the differentially expressed proteins were enriched by Bonferroni correction ($p < 0.05$). Furthermore, KOBAS software was applied to test the statistical enrichment of DEPs in KEGG pathway based annotation system KOBAS 2.0 (KOBAS, http://kobas.cbi.pku.edu.cn).

## Quantitative real-time PCR (RT-qPCR) analysis

RT-qPCR experiments used the same TAT samples with the proteomics. RNA samples were extracted according to the manufacturer's manual by Tiangen Kit (Tiangen Bio, China). Reverse transcription was performed by a Takara Ex Taq Kit (Takara Bio. Inc., Shiga, Japan). Genome-wide comparison of FTS and TTS selection signatures or association studies with the fat tailed phenotype have identified a number of candidate genes linked with the presence of a fat tail [2, 5, 9, 12–15, 17, 20–23, 25]. Therefore 24 proteins, based on their functions and expression level, were selected to test their gene expression levels in the present study (S1 Table). Among them COL1A1, SERPINC1 and ASPN were highly expressed in TTS vs. FTS, which were used as control. The primers for RT-qPCR are listed in S1 Table, which were designed by Primer 5.0 plus software. RT-qPCR reactions were run on the CFX 96 system (Bio-Rad, USA) with a 20 μl reaction volume, which contained 2 μl of cDNA template, 10 μl of 2 × SYBR Green Master Mix (RR420A, Takara, Dalian, China) and 0.5 μl of each primer (10 μmol/μl). The RT-qPCR reaction for each gene was performed with three biological replicates. Relative gene expression was normalized to the expression of Glyceraldehyde-3-Phosphate Dehydrogenase (*GAPDH*) and calculated with the $2^{-\Delta\Delta CT}$ method.

## Western blot, immunohistochemistry and immunofluorescence analysis

Western blot analysis was performed to validate the variation tendencies of DEPs identified by our proteomics approach. The protein bands were visualized by using an enhanced chemiluminescence ECL reagent (Thermo Fisher Scientific). Primary antibodies and their dilution ratio were followed, Anti-COL1A1 (1:10000, ab138492), Anti-ACSL1 (1:10000, ab177958), Anti-ERK2 (1:10000, ab32081) (Abcam, USA); HSD17β4 antibody (1:2000, 15116-1-AP), FABP4 antibody (1:1000, 12802-1-AP), PKA C-alpha antibody (1:1000, 55388-1-A), ATGL antibody (1:2000, 55190-1-AP) (Proteintech, USA), and β-actin (polyclonal antibody, 1:1000, Shenggong, China).

Immunohistochemistry staining was performed to map the distribution of FABP4 protein in the TAT according to a previously described protocol [35]. The tail adipocyte sections (thickness, 12 mm) were obtained in Tibetan sheep (TTS) and Hu sheep (FTS) and incubated overnight at 4˚C with the FABP4 antibody, which was diluted 1:1000 in PBS/1% BSA, followed by incubation for 1 h at room temperature with fluorochrome-conjugated secondary antibody anti-rabbit IgG A0208 (Biyuntian, China). We performed immunofluorescence to confirm this correlation, followed by incubating in the corresponding secondary antibodies. The Alexa-Fluor-488 conjugated mouse IgG (Bioss, China) was incubated for 1 h at room temperature and then diluted 1:200 in PBS/1% BSA.

## Statistical analysis

The cell diameters of adipocytes in different breeds were presented as the means ± SEM. One-way analysis of variance (ANOVA) with Tukey's posttest was used for statistical significance analysis with SPSS version 18.0 (SPSS, Inc. Chicago, IL, USA). The Kruskal-Wallis test with Dunn's Multiple Comparison test was used to detect statistically significant differences of gene and protein relative expressions in three FTS vs. two TTS by GraphPad Prism 5 (Prism Graph-Pad, San Diego, CA), and ANOVA followed by Student-Newman-Keuls post-test was used to determine exactly where this difference occurred. Relative expression was presented as the Mean ± SD. In the figures and tables, data shown without a common letter differ at a $p < 0.05$ significance level.

## Results

### The comparative analysis of adipocytes size for TAT between FTS and TTS

In this study, we observed the adipocytes morphology of TAT from five sheep breeds. The results showed that adipocytes in the three fat-tailed breeds were significantly larger than that in two thin-tailed breeds (S1 Fig). Additionally, the volumes of TAT in the three fat-tailed lambs were larger than the two thin-tailed lambs. These results suggested that TAT in FTS might be with higher efficiency for fat accumulation than that of TTS.

### Quantitative comparisons of the proteomes in FTS vs. TTS

The global protein expression profiles in the TATs of the five sheep breeds were revealed by the quantitative proteomic analysis according to the experimental design workflow (Fig 1B). In the tail or rump adipose tissues of five sheep, 2,303 (Kazakh), 2,137 (Lanzhou big fat-tail), 2,266 (Hu), 1,932 (Merino), and 1,941 (Tibetan) proteins were detected, respectively. A total of 3,400 proteins were successfully identified in these TATs, and 1,209 proteins were found to be overlapped in all the five tissues (Fig 1C). Furthermore, a label-free quantitative strategy was employed to compare the level of protein abundance at the different tail types, 804 DEPs were finally identified (Fig 2A and S1 File). To better understand the biological function of these DEPs, enrichment analysis was performed, and most of the DEPs were found to be enriched in lipid metabolic process, PPAR, and adaptive thermogenesis, these related proteins exhibiting a

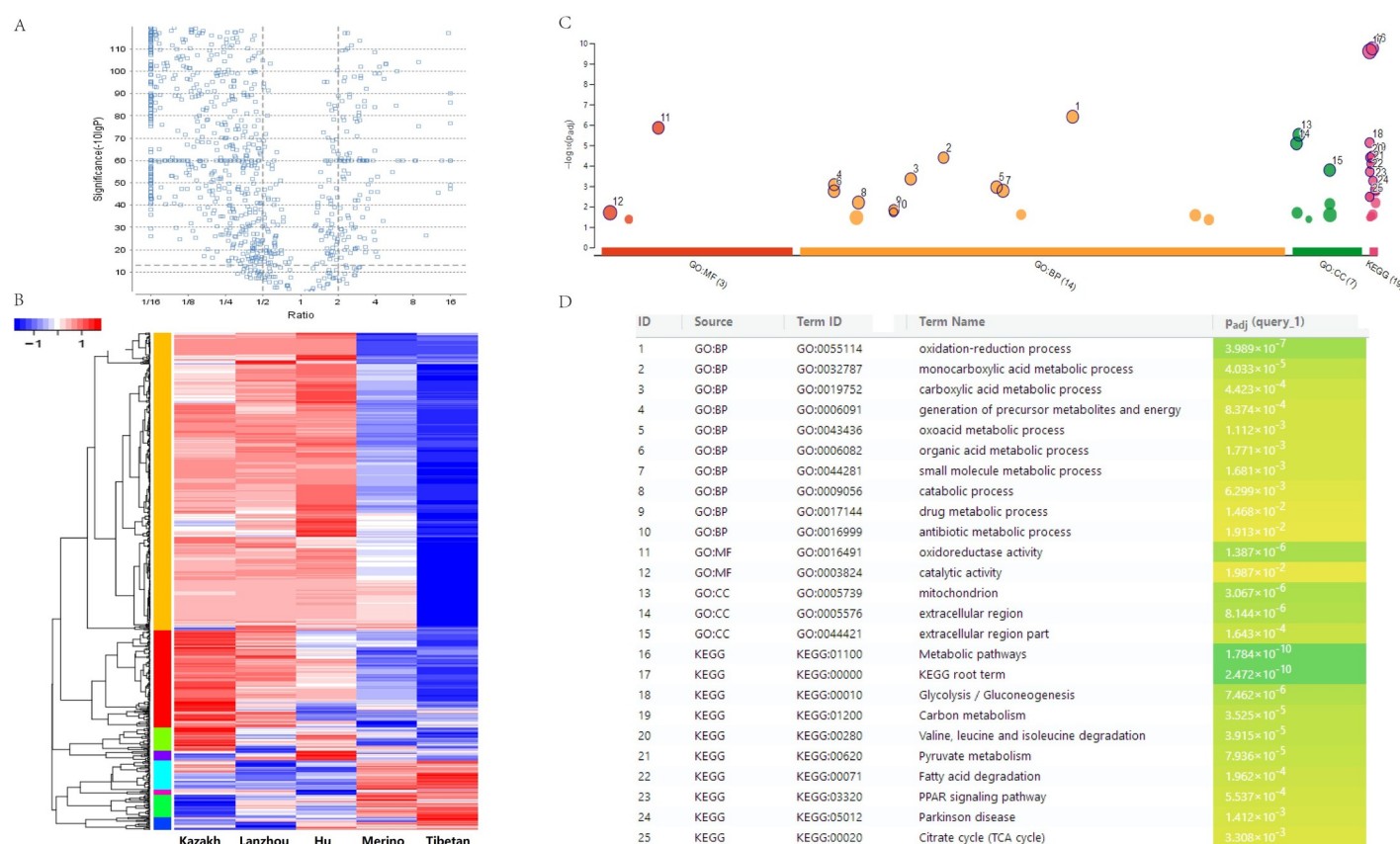

**Fig 2. Volcano plot, heat map analyses and GO enrichment of the differently expressed proteins.** (A) Volcano plot for DEPs. (B) All the differential proteins are clustered and visualized as a heat map. (C) and (D) The 804 differently expressed proteins were enriched based on GO categorization.

higher level of expression in FTS than in TTS, moreover, FABP4 as the highest protein expression level in all the 5 tested fat tissues, and adipokines and growth factor like ADIPOQ, MYDGF and TGFBI were found with different expression between FTS vs. TTS (S1 File). The heat-map was performed to investigate the expression profiles patterns of these DEPs, and results showed that three FTS were clustered together, and the two thin-tailed breeds were clustered together (Fig 2B). We also performed a clustering analysis on all 804 DEPs, and 8 clusters were observed, including two main clusters, each of which represented a distinct expression pattern, Cluster K1 and K2 containing 638 DEPs (79.35%), they represented a protein expression profile that were up-regulated in FTS but down-regulated in TTS. Cluster K5 and K7 included 83 (10.32%) proteins that were down-regulated in FTS but up-regulated in TTS (S2 Fig), and other four clusters for the DEPs may be particular with higher or lower expression profile in one breed. Many of the up-regulated proteins in the FTS, including HSL, PLIN1, PLIN4, ACSL1, FABP4, FABP5, which were considered as candidate proteins for fat metabolism and they can potentially involve in the generation of energy, adipogenesis and lipogenesis processes. Additionally, COL1A1, SERPINC1 and ASPN, involving in collagen fibril organization were validated highly expressed in TTS (S1 File).

## Biological processes and pathways analysis of the differentially expressed proteins

To determine the functional groups, all of the 804 statistically significant differences in protein abundance were conducted GO terms and KEGG pathway analysis. 14 biological progress were significant enrichment ($p < 0.05$), including 'oxidation-reduction process', 'carboxylic acid metabolic process', 'small molecule metabolic process' (Fig 2C and S2 File). Moreover the major enriched pathways analysis showed that 170 proteins significantly enriched in metabolic pathways ($p = 1.78 \times 10^{-10}$). In addition, several lipid metabolism related pathways were found, including fatty acid metabolism ($p = 5.78 \times 10^{-5}$), fatty acid degradation ($p = 1.96 \times 10^{-4}$), and fatty acid elongation ($p = 3.43 \times 10^{-2}$) were also significantly enriched, respectively (Fig 2D and S2 File). Furthermore, 19 proteins were found in the PPAR signaling pathway ($p = 5.53 \times 10^{-4}$) (Fig 3), including FABP4, FABP5, ACSL1, ACSL6, PLIN1, PLIN4, SCD, LPL, ACAA1, ADIPOQ. Therefore, these data showed that these pathways should contribute to tail fat deposition processes in sheep, and we focused on PPAR pathways for further analysis. We also conducted GO categories analysis for the up-regulated 638 proteins in FTS and the results were shown in S2 File, the most enriched GO term in FTS versus TTS was also 'oxidation-reduction process'. While the GO categories and KEGG pathway enrichment analysis for down-regulated 83 proteins (FTS vs. TTS) were found to be predominantly related to the 'blood coagulation', 'collagen fibril organization' and 'extracellular structure organization', all the GO terms and KEGG pathways with significant enrichment were listed in S2 File.

## Verification of the DEPs' expression levels

To verify the accuracy of the proteomics data, 24 proteins were selected to determine their gene expression levels in the five samples (Figs 4B and S3). RT-qPCR results showed that almost all the 24 gene expression patterns were consistent with the proteomics abundance changes in the five Chinese native sheep breeds, illustrating that the gene expression level was consistent with protein expression in TAT. Moreover, COL1A1, FABP4, ACSL1, ATGL, HSD17β4, ERK2, and PRKACA were selected for western blot analyses to confirm the MS observations. All the results in our study showed that the protein expression were consistent with the data of proteomics, and the expression of FABP4, ACSL1, ATGL, HSD17β4 in fat-

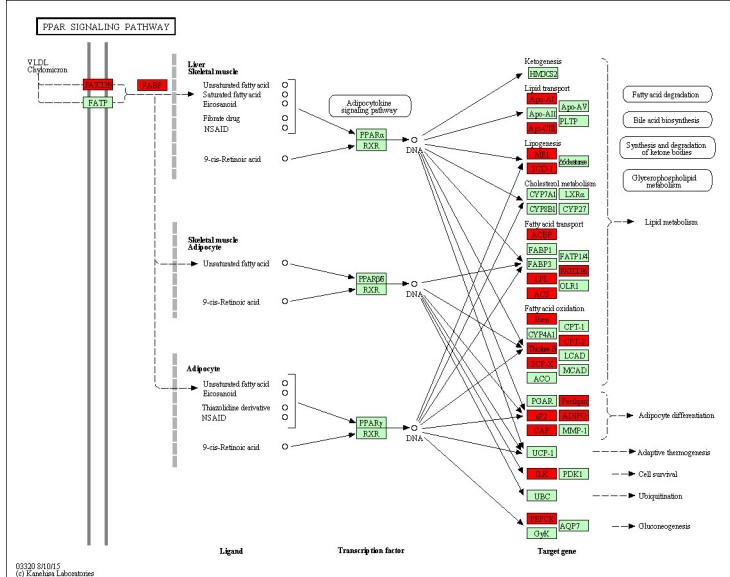

**Fig 3. An overview map of the PPAR pathway.** The identified differently expressed proteins involved in the PPAR were the red mark, and enriched based on KEGG pathways.

tailed were higher than thin-tailed (Fig 4A and 4B), suggesting that the expression profile determined by the present comparative proteomic approach was reliable.

### The results of immunohistochemistry and immunofluorescence of FABP4

*FABP4* gene expression in fat tissues can be considered as an important index of fat deposition. Immunohistochemistry and immunofluorescence were performed to validate the expression of FABP4 in the tail fat tissues. Our results showed that FABP4 were specifically expressed in the capillary wall of TAT, and its expression level in the fat-tailed lamb was higher than that in the TTS (Fig 5). FABP4 highly expressed in the capillary wall may be beneficial to the cell proliferation and transportation for the fatty acids from the blood to the adipocytes.

### Discussion

It has always been of interest to reveal the genetics and molecular regulation mechanisms for tail fat deposition in FTS. The proteomic-based approach can be a practical and efficient way to exploit protein targets associated with tail/rump fat accumulation and provide an opportunity to better understand the mechanism of tail fat deposition in sheep. To our best knowledge, the present research is the first study to utilize a label-free quantitative proteomic analysis technique to determine the differences in protein levels in TATs between fat- and thin-tailed lambs. Our results show that the differentially expressed proteins involved in PPAR signaling pathway might regulate the adipocyte differentiation and lipid metabolism in tail/rump of sheep, especially the high expression of FABP4 in TAT of FTS, which will be valuable for future studies on tail fat deposition.

Sheep TAT is a special organ for energy storage, endocrine regulation and energy metabolism. Our quantitative proteomic analysis has characterized protein expression profiles of tail fat tissues in five different tail type sheep breeds, which can provide a comprehensive understanding of the molecular regulating mechanism underlying the fat tail phenotype in sheep. In total, 3,400 proteins were identified in all the five tail or rump adipose tissues, compare with

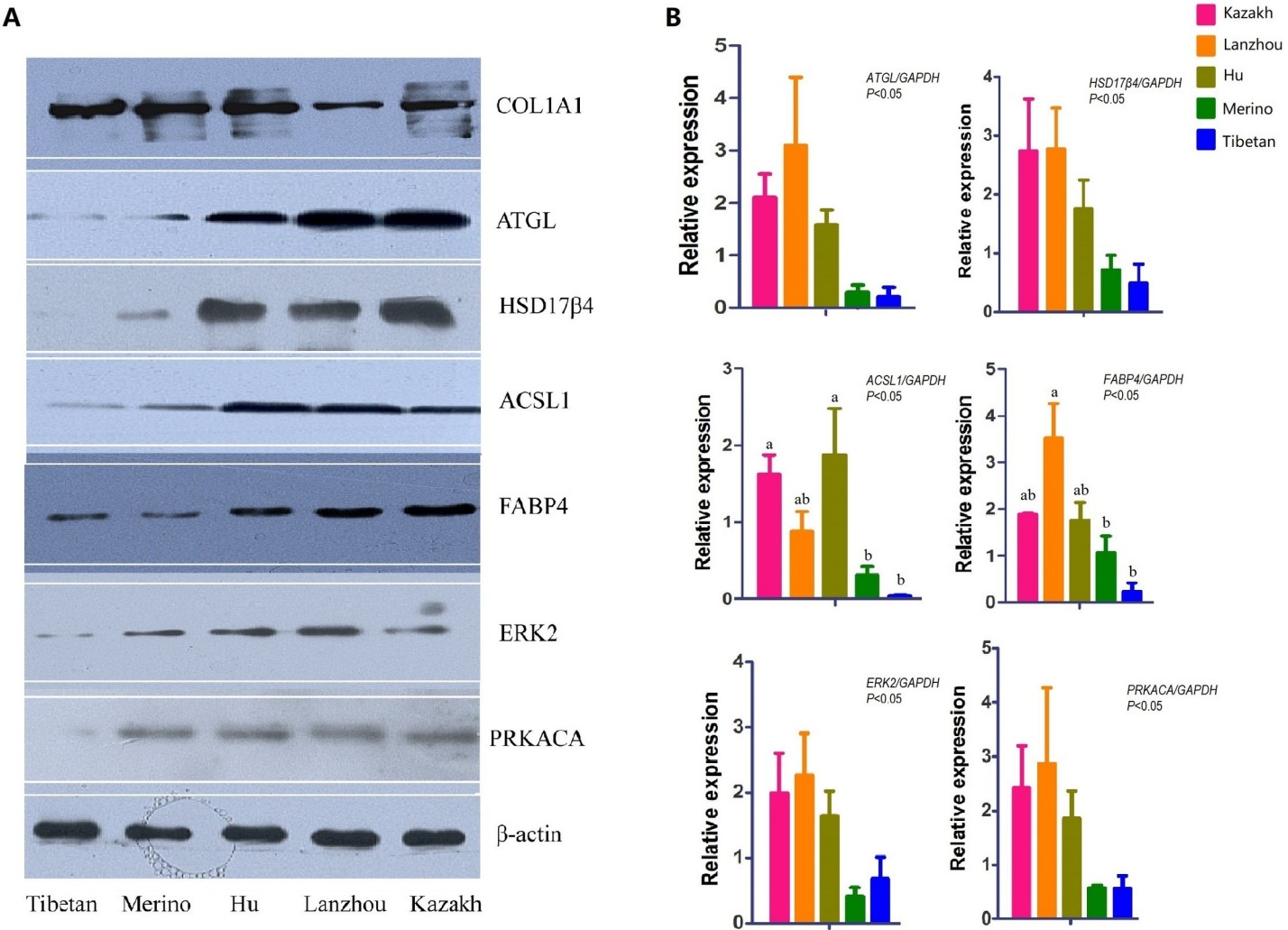

**Fig 4. Western blot and RT-qPCR assay of ATGL, ACSL1, HSD17β4, FABP4, ERK2 and PRKACA in the tail/rump adipose tissues of five sheep breeds.** (A) Western blot bands of COL1A1, ATGL, ACSL1, HSD17β4, FABP4, ERK2, PRKACA. (B) Statistical data of the mRNA of their gene expression. *GAPDH* was used as reference control. The *p* values of the relative gene expressions in five breeds were 0.0310 (*ATGL*), 0.0477 (*HSD17β4*), 0.0171 (*ACSL1*), 0.0237 (*FABP4*), 0.0337 (*ERK2*) and 0.0347 (*PRKACA*). Differences with *p* values <0.05 were considered to be statistically significant, bars without a common letter differ, $P < 0.05$.

the previous reports, there was only a similar study had identified 1,610 proteins in rump adipose tissues of Kazakh sheep by using iTRAQ-labelling proteomics-based approach [40]. In addition, the cluster analysis of the proteins expression profile showed that three FTS were clustered together, and two TTS were clustered together, which revealed that the FTS breeds exhibit a greater similarity than the other TTS breeds. However, there was a little variation in each breed, which can be attributed to a low level of individual variation. A total of 804 DEPs were identified in three fat tailed lambs vs. two thin tailed lambs including Apo-AI, LPL, LEP, and ADIPOQ. To comprehensively analyze the genetic molecular mechanism for tail fat deposition in different sheep breeds, the differentially abundant proteins were subsequently subjected to GO annotation and KEGG enrichment analysis. Functional enrichment analysis suggested that most of the DEGs were directly or indirectly involved in the oxidation-reduction process, which as the main metabolic process was observed by the previous studies [11, 28]. Most of the significantly enriched GO terms and KEGG pathways of up-regulated DEPs

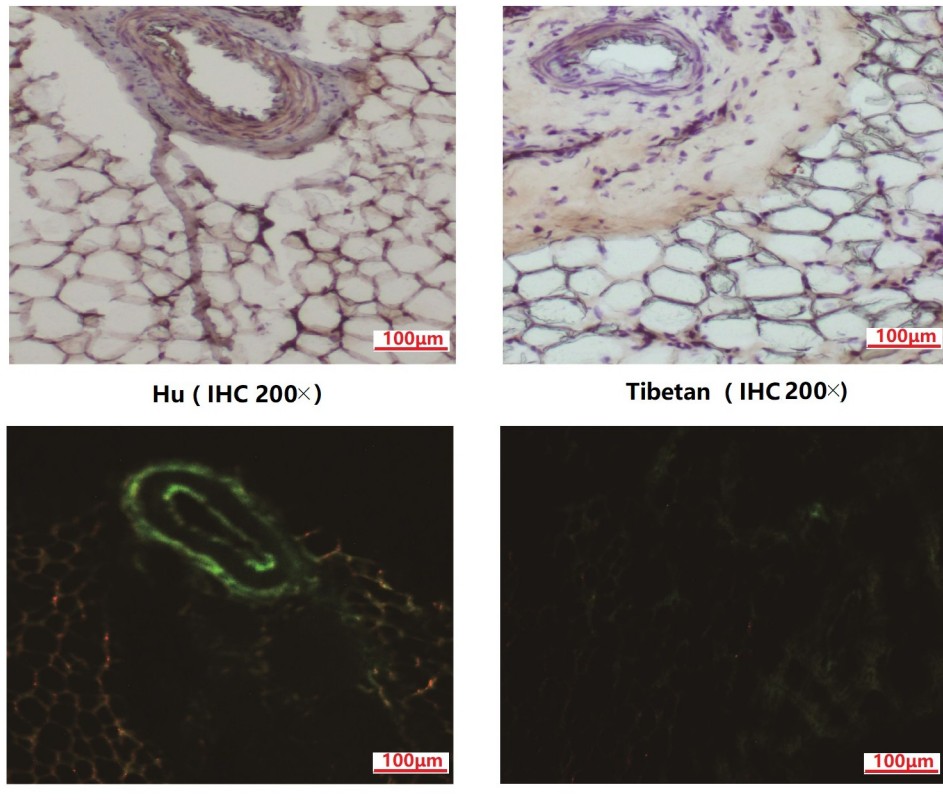

**Fig 5. IHC and immunofluorescence analysis of FABP4 in the tail fat tissues between fat- and thin- tailed sheep.**
IHC and immunofluorescence results demonstrated the localization of the differently expressed FABP4 in TAT of Hu
sheep (FTS) and Tibetan sheep (TTS).

in the fat tailed breeds were directly related to 'metabolic process' and 'catabolic process', and
'lipid metabolism'. This observation may provide support to the previous gene expression-
based studies, which have found the DEGs were related to lipid metabolism in TAT of other
sheep breeds [11, 26, 33]. Besides, Wang et al. [41] proposed that many genes involved in adi-
pogenesis, fatty acid biosynthesis, and lipid metabolism were identified for fat deposits in the
tail/rump region. However, Miao et al. [28] showed a predominant down-regulation of mas-
sive metabolic processes were necessary for tail fat deposition. We noticed that most of the pre-
vious studies were selected adult sheep to collect TAT, thus the results of these studies may
remain inconclusive, because the tail fat deposition could be affected by the diet nutrient levels
and the seasons. Our study chose TAT from the lambs could eliminate the interferences. Thus,
results of the current study provide further evidence that these DEPs enriched in metabolic
process biological processes could contribute fat deposition in tail tissue.

The synthesis and degradation of fatty acids determining deposition or utilization of depot
fat in adipose tissue [42]. There are several functional candidate genes associated with fat depo-
sition and adipogenesis in FTS [3], including ACSL1, ACACA, ACLY, FASN, ATGL and HSL,
which also were validated and considered as the key factors determining lipid metabolism in
fatty tissues [26, 41, 43]. ACSL1 is essential for the synthesis of the progress of adipogenic dif-
ferentiation culminating in triacylglycerol synthesis and storage [44]. In our data, we noticed
that ACSL1, ACACA, ACLY, and FASN were up-regulated in the three FTS breeds, which are

in agreement with the previous transcriptome analysis on other sheep breeds [41, 43]. Nevertheless, ATGL and HSL are the two main lipases and are responsible for 95% of the lipolysis in adipose tissue [45]. Interestingly, in our observations, ATGL and HSL were observed with high expressions in TAT of FTS. Adipose tissue can be a major source of metabolic fuel, and a number of enzymes are involved in fat accumulation in adipocytes. The results suggested that these up-regulated proteins might play key roles in regulating fatty acids transportation, fatty acid synthesis and formation of adipose tissue.

Fat deposition is also a process for the differentiation of adipocytes. The increase of adipocyte mass and adipocyte number are both vital to adipogenesis [46, 47]. A recent study has found tail adipose to be more efficient for fat deposition than other fat tissues in FTS [27]. Therefore, we hypothesized that the ability of fat deposition in tail adipose of fat tailed/rumed sheep is more efficient than TTS. Similarly, Bakhtiarizadeh et al. [26] suggested that the regulation of adipocyte diferentiation might be different in fat- (Lori-Bakhtiari) and thin-tailed (Zel) Iranian sheep breeds. Furthermore, more adipocytes were found in the TAT of FTS than in TTS according to the size and volume, in addition many up-regulated proteins associated with adipogenesis and proliferation were highly expressed in TAT of FTS, especially, FABP4, HSD17β4 and PLIN1 showed higher levels in FTS than TTS. RT-qPCR and western blot were used to validate their gene and protein expression level. PLIN1, HSD17β4 and FABP4 could participate in the control of adipocyte differentiation based on the previous research findings [48, 49]. Therefore, the data and analysis presented in our study suggest that TAT of fat-tailed lambs have a higher efficiency of fat accumulation. Furthermore, ERK2 and PRKACA, these two key kinases are important for most of the metabolic process related pathways, which were detected with high expression in the TAT of FTS by western blot. PRKACA is one of the catalytic subunits of protein kinase A, and PKA can phosphorylate and activate HSL, regulating lipolysis [50]. Thus, it is likely that ERK2 and PRKACA could participate in proliferation and differentiation of tail adipocytes. Based on this, our results provide new light on the underlying tail fat deposition mechanisms at the molecular level.

High amount of *FABP4* in fat-tail can play key roles in regulating differentiation of adipocytes, and recent studies had shown that significantly higher expression of FABP4 in TAT of FTS vs. TTS [11, 30, 51]. The present study confirmed FABP4 were newly identified highly expressed at the inner and outer membrane of the capillary wall or around the blood vessel in TAT of all the three FTS. Although previous study had reported this particular candidate gene was very important for fat deposition, but we fist proved it was especially expressed at the inner and outer membrane of the capillary wall or around the blood vessel in our observation, so the evidence support our speculation that high expression of FABP4 may regulate the function of the differentiation of pre-adipocyte and fatty acid transport from the blood to the adipocytes leading to tail fat deposition. FABP4 also can induced phosphor-ERK signal, and Bakhtiarizadeh et al. [26] had proved that MAPK signaling pathway is an important component contributing to modulation of gene network of fat deposition in tail tissue. As the main kinase of MAPK signaling pathway, the proliferation and adipogenic differentiation of adipostromal cells will be influenced by that of ERK1/2 activation [52]. In this present study, ERK2 as upstream elements were found in the FTS, this supports the previous observations, suggested that ERK2 may contribute to activation of adipogenic and lipogenic. Taken altogether, these data indicate that at least in part, the FABP4 plays a key role in sheep tail fat deposition, and FABP4 may enhance the cell proliferation in the TAT. Hence, it appears that the high proliferative ability for adipocytes in lambs of FTS is more important for fat deposition in the late stage. All these evidences support our speculation that TAT in FTS would have a rapid preadipocyte proliferation and differentiation rather than TTS. To validate this hypothesis, further

investigation on the candidate genes and pathways may help to understand adipocyte proliferation and differentiation of TATs both in FTS and TTS.

To obtain further insight into the molecular events controlling fat accumulation in TAT, a pathway-based analysis allows further understanding of the biological functions of DEPs revealed in the current research. Based on a pathway-enrichment analysis, metabolic pathways, PPAR signaling pathway and fatty acid metabolism process were identified with significantly enriched. PPAR signaling pathway plays key roles in regulating cellular differentiation and cell proliferation [11], which was identified as the frequently used pathway related to the fat deposition [26, 53]. PPAR signaling pathway genes (including *FABP4*, *FABP5*, *ACSL1*, *ACSL6*, *PLIN1*, *PLIN4*, *SCD*, *LPL*, *ACAA1*, and *ADIPOQ*) were reported as up-regulated DEGs in TAT of other FTS in comparison with TTS breed previously [10, 11, 27–29, 41, 43, 51]. All these genes also have been investigated by RT-qPCR in the present study, which were consistent with these previous studies. Therefore, based on these results, these up-regulated genes in PPAR signaling pathway might contribute to fat deposition in the tail or rump of sheep.

## Conclusion

In summary, in order to investigate the tail fat deposition in sheep, an attempt was done to identify the changes in protein levels for five Chinese typical sheep breeds with different tail types. We identified a total of 3,400 proteins in all the TATs of five sheep. Of which, 804 were DEPs in three fat tailed lambs vs. two thin tailed lambs, among them FABP4, ACSL1, ACACA, ACLY, FASN, and HSD17β4 were highly expressed in TAT of FTS and they may play important roles in tail fat accumulation. Moreover, the PPAR signaling pathway genes (*FABP4*, *FABP5*, *ACSL1*, *ACSL6*, *PLIN1*, *PLIN4*, *SCD*, *LPL*, *ACAA1*, *ADIPOQ*, etc) were previously reported to be correlated with fat tailed formation, which were also further validated in current study, and these candidate genes could be further considered as genetic biomarkers for fat tailed trait. Therefore, further works are required to confirm and refine our results and investigate the role of these specific genes. Notwithstanding, the present findings offer new insights into the tail fat accumulation in sheep, and provide an important theoretical basis for future clinical and basic research on this trait.

## Supporting information

**S1 Fig. Adipocyte diameter comparative analysis of TAT in the five breeds.** (A) H&E staining for the TAT; (B) Cell diameter analysis. Cell diameter unit was μm, bars without a common letter differ, $P < 0.05$.
(DOC)

**S2 Fig. The eight cluster of the DEPs.** PA means protein areas. K represent Kazakh sheep, L represent Lanzhou big tailed sheep, H represent Hu sheep, M represent Alpine Merino sheep, T represent Tibetan sheep.
(DOC)

**S3 Fig. RT-qPCR verification quantitative proteomic of DEPs.** 18 proteins were selected, with GAPDH as the internal reference gene. The *P* values of relative expression of gene and its protein: 0.1526(*AACS*) and 0.0087 (AACS); 0.0192 (*ACACA*) and 0.0086 (ACACA); 0.0337 (*ACADVL*) and 0.0091 (ACADVL); 0.0171 (*ACSS2*) and 0.0087 (ACSS2); 0.0197 (*ADIPOQ*) and 0.0101 (ADIPOQ); 0.0477 (*ADIRF*) and 0.0117 (ADIRF); 0.0337 (*ASPN*) and 0.0101 (ASPN); 0.0302 (*ELOVL6*) and 0.0086 (ELOVL6); 0.0289 (*FASN*) and 0.0087 (FASN); 0.0377 (*FABP5*) and 0.0120 (FABP5); 0.1060 (*HACD2*) and 0.0087 (HACD2); 0.0192 (*HADH*) and 0.0112 (HADH); 0.0458 (*HSL*) and 0.0091 (HSL); 0.0258 (*PLIN1*) and 0.0111 (PLIN1); 0.0410

(*PLIN4*) and 0.0110 (PLIN4); 0.0310 (*NDRG2*) and 0.0091 (NDRG2); 0.0452 (*TMEM120A*) and 0.0129 (TMEM120A); 0.3753 (*SERPINC1*) and 0.0117 (SERPINC1). Differences with *p* values <0.05 were considered to be statistically significant.
(DOC)

**S1 Table. The primer sequence for RT-qPCR.**
(DOC)

**S1 File. All the 804 differentially expressed proteins in tail adipose tissue of all the five breeds.**
(XLS)

**S2 File. GO and pathways analysis of the differentially expressed proteins.**
(CSV)

**S1 Raw image.**
(PDF)

## Acknowledgments

The authors would like to thank Bin Han, Yu Fang and Mao Feng for their kind expert technical assistance.

## Author Contributions

**Data curation:** Tingting Guo, Jianbin Liu, Chune Niu.

**Formal analysis:** Zengkui Lu, Chao Yuan.

**Methodology:** Chao Yuan.

**Resources:** Yaojing Yue.

**Validation:** Zengkui Lu, Bohui Yang.

**Visualization:** Bohui Yang.

**Writing – original draft:** Jilong Han, Tingting Guo, Min Yang, Bohui Yang.

**Writing – review & editing:** Yaojing Yue, Min Yang, Bohui Yang.

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
