## [Decision Letter · Decision Letter 0]

9 Nov 2020

PONE-D-20-24379

Quantitative proteomic analysis identified differentially expressed proteins with tail/rump fat deposition in Chinese thin- and fat-tailed lambs

PLOS ONE

Dear Dr. Yang,

Thank you for submitting your manuscript to PLOS ONE. After careful consideration, we feel that it has merit but does not fully meet PLOS ONE’s publication criteria as it currently stands. Therefore, we invite you to submit a revised version of the manuscript that addresses the points raised during the review process.

We look forward to receiving your revised manuscript.

Kind regards,

Livia D'Angelo

Academic Editor

PLOS ONE

Journal Requirements:

"This research was funded by the National Natural Science 379 Foundation of China,

 grant numbers 31702092 and 31802045; Modern China Wool & Cashmere Technology Research

System, grant number CARS-39-02; Agricultural Science and Technology Innovation Program for

Genetic Resource and Breeding of Fine-Wool Sheep, CAAS, grant number ASTIP-2015-LIHPS, and

Engineering Research Center of Sheep and Goat Breeding, CAAS; Special Funds for Basic Research of

Young Scholars of Shihezi University, grant numbers CXBJ201903, CXRC201808, RCSX2018B10

and RCSX2018B11."

Reviewers' comments:

Reviewer's Responses to Questions

**Comments to the Author**

1. Is the manuscript technically sound, and do the data support the conclusions?

Reviewer #1: Yes

Reviewer #2: Yes

Reviewer #3: Yes

2. Has the statistical analysis been performed appropriately and rigorously? 

Reviewer #1: Yes

Reviewer #2: Yes

Reviewer #3: Yes

3. Have the authors made all data underlying the findings in their manuscript fully available?

Reviewer #1: Yes

Reviewer #2: Yes

Reviewer #3: Yes

4. Is the manuscript presented in an intelligible fashion and written in standard English?

Reviewer #1: No

Reviewer #2: Yes

Reviewer #3: Yes

5. Review Comments to the Author

Reviewer #1: In this manuscript, several concerns need to be addressed to fit for publication as follows:

General comments:

1. There is a problem in using the abbreviations throughout the manuscript especially in the abstract. The abbreviation must be introduced upon the first mentioning of the full term followed by its abbreviation in parentheses: From then on, the abbreviation must be used exclusively and throughout. E.g. Fat-tailed sheep (FTS) in line 46 then the full term has repeated again many times throughout the manuscript lines 89, 201, 208, 211…etc.

2. The manuscript contains many grammatical, typographic (especially capitalized letters e.g. line 39 Domesticated; line 80 Five), and styling errors. English editing of the manuscript is highly recommended.

Specific comments:

1. Abstract:

- It needs to more informative and carefully arranged as a brief summary of the background, purpose, methodology (more details are needed), results, and conclusion.

- Line 31: there is no need to give an abbreviation for peroxisome proliferator-activated receptor (PPAR) as it has not been repeated again in the abstract.

2. Introduction: the novelty of the study compared to the other earlier proteomics-based studies (Ref: 6, 22-27) need to be more clarified before the aim.

3. Material and methods: Please, add the reference for the protocol of the adipose tissue staining method.

4. Results:

- The significant letters or symbols denoting the significant difference among the different breeds are missed in Fig.2b. Please, add. Also, the y-axis should have the name of the analyzed gene and housekeeping gene (e.g. relative mRNA expression ATGL/GAPDH) and remove the name of the gene from the x-axis.

Reviewer #2: Comments on manuscript PONE-D-20-24379:

Abstract:

Some information id required concerning the following:

1- Clearly indicate the aim of the study .

2- Indicate the breeds of sheep in this study.

Methodology:

Some information id required concerning the following:

1- The components of the diet used in the current study and its chemical composition.

2- Lambs weights averages should be reported at the start and end of this study.

3- The period of study and the date of its work must be mentioned.

4- Some study parameters (adipose tissue staining, quantitative proteomic analysis and bioinformatics analysis ) should be measured based on scientific references.

Discussion:

Unsatisfied in general. As compared with his results the authors shows the agreement or disagreement with literature without pay enough attention to discuss the reasons.

Reviewer #3: dear Authors,

in future research in this field I suggest to involve correlation between up regulated proteins in adipose tissue in tail and adipose tissue in muscle, that will assist researcher to understand if they can increase marbling features especially in these sheep.

6. PLOS authors have the option to publish the peer review history of their article (what does this mean?). If published, this will include your full peer review and any attached files.

Reviewer #1: No

Reviewer #2: **Yes: **Assist. Prof. Dr. Khalid Chillab Kridie Al-Salhie

Reviewer #3: No

---

## [Author Response · Author response to Decision Letter 0]

4 Jan 2021

Dear Editors and Reviewers:

Thank you for your letter and for the reviewers’ comments concerning our manuscript entitled “Quantitative proteomic analysis identified differentially expressed proteins with tail/rump fat deposition in Chinese thin- and fat-tailed lambs”. Those comments are all valuable and very helpful for revising and improving our paper. Based on reviewer’s comments, we have made extensive revisions to our previous draft. According to your suggestions, the introduction and discussion are rewritten in our manuscript, and the revisions in the text are shown using red highlight for additions. We addressed all the points raised by the reviewer as summarized below.

We firstly updated the manuscript by adjusting the style and references, so we hope the manuscript has meet the PLOS ONE's style requirements according to the templates. In additional, the original uncropped and unadjusted western blot image data are list in the supporting information (S1_raw_image).

Comments on manuscript PONE-D-20-24379 entitled (Quantitative proteomic analysis identified differentially expressed proteins with tail/rump fat deposition in Chinese thin- and fat-tailed lambs).

Title:

It is good and appropriate for the article.

Abstract:

Some information id required concerning the following:

1-Clearly indicate the aim of the study.

Response: Thank you for your good advices. We rewrite the aim of the abstract, as the follow: Tail adipose as one of the important functional tissues can enhance hazardous environments tolerance for sheep. The objective of this study was to gain insight into the underlying development mechanisms of this trait.

2-Indicate the breeds of sheep in this study.

Response: Thanks for your suggestion, we have rewritten this sentence: A quantitative analysis of protein abundance in ovine tail/rump adipose tissue was performed between Chinese local fat- (Kazakh, Hu and Lanzhou) and thin-tailed (Alpine Merino, Tibetan) sheep in the present study by using lable-free approach. 

Introduction:

The introduction is comprehensive and covers the topic of the study.

Methodology:

Some information id required concerning the following:

1-The components of the diet used in the current study and its chemical composition.

Response: Thanks for the reviewer’s good advice, we agree that the components of the diet used in the current study and its chemical composition would be very important to understand the genetic effect and effects on feeding environment. In addition, tail fat deposition would be influenced by different nutrient levels. However, at this point all the lambs selected in this study were not weaning, and the lambs had been located indoors with their dams during lactation, and reared under similar conditions with supplemental cracked corn and dry alfalfa and allowed free access to feed and water. Thus, there will be difficult to analyze the components of their diet now, which contains milk, energy and protein feed. We believe all the lambs can be raised in an ideal environment, the different of tail fat deposition would be caused by genetics as the main reason, and the different amount of fat may be associated with the genetic variation or different regulation of gene expression between different breeds.

Lambs weights averages should be reported at the start and end of this study.

Response: Thanks for your suggestion, all the lambs selected at birth and sacrificed with 13-15 kg live weight in this study. 

The period of study and the date of its work must be mentioned.

Response: Thanks for the reviewer’s good suggestion, the lambs were selected at birth, and they were sacrificed (distributed at periods of April 20, 2014-April 28, 2014; 13-15 kg live weight; 40±10 days on average)

Some study parameters (adipose tissue staining, quantitative proteomic analysis and bioinformatics analysis ) should be measured based on scientific references.

Response: We are sorry for the unclear description in the original manuscript, all the adipose tissue staining, quantitative proteomic analysis and bioinformatics analysis methods used in this study were added the references. 

The adipose tissue staining was performed by the method described by Ruoss [35]. 

Ultimate nano HPLC 1000 system coupled with QExactive quadrupole-orbitrap mass spectrometer (Thermo Scientific, Bremen, Germany) was utilized for label-free analysis according to the previously described method [37]

The generated raw data from label-free LC-MS/MS were further examined by PEAKS software (version 7.5, Bioinformatics Solutions, Waterloo, Canada) as our previously described method [38].

Gene Ontology (GO) enrichment analysis of DEPs was implemented using g:Profiler (http://biit.cs.ut.ee/gprofiler/) as described previously [39]

Results:

The results are good and illustrated the figures in detail.

Discussion:

Unsatisfied in general. As compared with his results the authors shows the agreement or disagreement with literature without pay enough attention to discuss the reasons.

Response: We are sorry for our inappropriate sentences in discussion, therefore, we try our best to improve the quantity of this part, adding more context to discuss the reasons why the results agree or disagree with the previous literatures in the revised manuscript with red color.

Conclusions:

It is good .It is drawn appropriately based on the data presented.

References:

1-Some references are very old.

2-Some references need to write total pages.

3-This reference (Zhou GX, Wang XL, Yuan C, Kang DJ, Xu XC, et al.) without Vol., No and total pages.

Response: We want to thank reviewer for constructive advice about the reference. The reference was exported by Endnote, so we check the total pages of all the references at this time. According to your good suggestion, we also supplement some recent papers in the manuscript, some old references were replaced, after carefully checked, now we ensure the Vol., No and pages for the references were listed.

Response to comments (Reviewer):

Special thanks to the three reviewers for their evaluation of this manuscript and for their constructive comments regarding strengthening it, which have been accommodated appropriately. As the reviewer's good advice, we tried our best to improve the manuscript and made some changes in the manuscript. These changes will not influence the content and framework of the paper. Here we did not list all the changes but marked in revised manuscript. We appreciate for reviewers’ warm work earnestly, and hope that the correction will meet with approval.

We believe that the manuscript has been greatly strengthened by the critiques of the reviewers and hope that both you and the reviewers will now find the paper suitable for publication.

Reviewer #1: In this manuscript, several concerns need to be addressed to fit for publication as follows:

General comments:

1. There is a problem in using the abbreviations throughout the manuscript especially in the abstract. The abbreviation must be introduced upon the first mentioning of the full term followed by its abbreviation in parentheses: From then on, the abbreviation must be used exclusively and throughout. E.g. Fat-tailed sheep (FTS) in line 46 then the full term has repeated again many times throughout the manuscript lines 89, 201, 208, 211…etc.

Response: We are very sorry for our expression error, thus we have revised the sentence accordingly. All the abbreviations used in this manuscript: Fat-tailed sheep (FTS); thin-tailed sheep (TTS) and tail adipose tissue (TAT) were replaced the full term in the manuscript.

2. The manuscript contains many grammatical, typographic (especially capitalized letters e.g. line 39 Domesticated; line 80 Five), and styling errors. English editing of the manuscript is highly recommended.

Response: We are sorry for our inappropriate sentences, the capitalized letters (e.g. line 39 Domesticated; line 80 Five) were modified. Finally, we try to improve the grammar and expression in the English writing of the manuscript.

Specific comments:

1. Abstract:

- It needs to more informative and carefully arranged as a brief summary of the background, purpose, methodology (more details are needed), results, and conclusion.

Response: As the reviewer's good advice, we rewrite the background, purpose, methodology in abstract as follows: Tail adipose as one of the important functional tissues can enhance hazardous environments tolerance for sheep. The objective of this study was to gain insight into the underlying development mechanisms of this trait. A quantitative analysis of protein abundance in ovine tail/rump adipose tissue was performed between Chinese local fat- (Kazakh, Hu and Lanzhou) and thin-tailed (Alpine Merino, Tibetan) sheep in the present study by using lable-free approach.

 - Line 31: there is no need to give an abbreviation for peroxisome proliferator-activated receptor (PPAR) as it has not been repeated again in the abstract.

Response: We are sorry for our inappropriate sentences; the (PPAR) was deleted in this part.

2. Introduction: the novelty of the study compared to the other earlier proteomics-based studies (Ref: 6, 22-27) need to be more clarified before the aim.

Response: We are sorry for the unclear description. We have re-written this part according to the reviewer’s suggestion: some candidate genes were proved to be associated with the tail phenotype of sheep, including PPP2CA, EBP, PPP1CC, PDGFD, BMP2, VGR and VNRT [7, 13-24], among them, PDGFD has the strongest selection signal in different tailed sheep worldwide, indicating PDGFD may be involved in fat deposition in sheep tail [23, 24]. Nevertheless, the molecular mechanism underlying fat-tail development remains to be elucidated in sheep.

3. Material and methods: Please, add the reference for the protocol of the adipose tissue staining method.

Response: According the reviewer’s suggestions, we add the reference for the protocol of the adipose tissue staining method in the revised manuscript: The adipose tissue staining was performed by the method described by Ruoss [35].

4. Results:

- The significant letters or symbols denoting the significant difference among the different breeds are missed in Fig.2b. Please, add. Also, the y-axis should have the name of the analyzed gene and housekeeping gene (e.g. relative mRNA expression ATGL/GAPDH) and remove the name of the gene from the x-axis.

Response: Thank you for your good advices. It is our negligence that we just list the P value of each gene in the figure legend. We use Kruskal-Wallis test with Dunn's Multiple Comparison test to detect statistically significant differences of gene relative expressions in three FTS and two TTS by GraphPad Prism 5, meanwhile, ANOVA with Tukey’s Multiple Comparision test was used to determine exactly where this difference occurred. GAPDH was used as reference control to count the gene expression, finally we add the significant letters in Fig.4b, and data shown without a common letter differ at a p < 0.05 significance level.

Reviewer #2: Comments on manuscript PONE-D-20-24379:

Abstract:

Some information id required concerning the following:

1- Clearly indicate the aim of the study.

Response: The aim of the study in abstract was improved according to the reviewer’s suggestion. Tail adipose as one of the important functional tissues can enhance hazardous environments tolerance for sheep. The objective of this study was to gain insight into the underlying development mechanisms of this trait.

2- Indicate the breeds of sheep in this study.

Response: All the five breeds of sheep used in this study were list: A quantitative analysis of protein abundance in ovine tail/rump adipose tissue was performed between Chinese local fat- (Kazakh, Hu and Lanzhou) and thin-tailed (Alpine Merino, Tibetan) sheep in the present study by using lable-free approach.. Methodology:

Some information id required concerning the following:

1- The components of the diet used in the current study and its chemical composition.

Response: Thanks for the reviewer’s good advice, we agree that the components of the diet used in the current study and its chemical composition would be very important to understand the genetic effect and effects of feeding environment. In addition, tail fat deposition would be influenced by different nutrient levels. However, at this point all the lambs selected in this study were not weaning, and the lambs had been located indoors with their dams during lactation, and reared under similar conditions with supplemental cracked corn and dry alfalfa and allowed free access to feed and water. Thus, there will be difficult to analyze the components of their diet now, which contains milk, energy and protein feed. We believe all the lambs can be raised in an ideal environment, the different of tail fat deposition would be caused by genetics, and the different amount of fat may be associated with the genetic variation or different regulation of gene expression between different breeds.

2- Lambs weights averages should be reported at the start and end of this study.

Response: Thanks for your suggestion, all the lambs selected at birth and sacrificed with 13-15 kg live weight in this study.

3- The period of study and the date of its work must be mentioned.

Response: We are sorry for our unclear report, the lambs were selected at birth, and then lambs were located indoors with their dams during lactation, reared under similar conditions. All the lambs were slaughter to collect the adipose tissues at periods of April 20, 2014-April 28, 2014.

4- Some study parameters (adipose tissue staining, quantitative proteomic analysis and bioinformatics analysis ) should be measured based on scientific references.

Response: We are sorry for the unclear description in the original manuscript, the adipose tissue staining, quantitative proteomic analysis and bioinformatics analysis methods used in this study were added the references. 

The adipose tissue staining was performed by the method described by Ruoss [35]. 

Ultimate nano HPLC 1000 system coupled with QExactive quadrupole-orbitrap mass spectrometer (Thermo Scientific, Bremen, Germany) was utilized for label-free analysis according to the previously described method [37]

The generated raw data from label-free LC-MS/MS were further examined by PEAKS software (version 7.5, Bioinformatics Solutions, Waterloo, Canada) as our previously described method [38].

Gene Ontology (GO) enrichment analysis of DEPs was implemented using g:Profiler (http://biit.cs.ut.ee/gprofiler/) as described previously [39]

Discussion:

Unsatisfied in general. As compared with his results the authors shows the agreement or disagreement with literature without pay enough attention to discuss the reasons.

Response: Thanks for the reviewer’s good suggestion, we rewrite this part and discuss the reasons why the results agree or disagree with the previous literatures in the revised manuscript with red color.

Reviewer #3: dear Authors,

in future research in this field I suggest to involve correlation between up regulated proteins in adipose tissue in tail and adipose tissue in muscle, that will assist researcher to understand if they can increase marbling features especially in these sheep.

Response: Thanks for the reviewer’s kind advice. 

As we all known marbling features in livestock is so important, I think some of the up regulated proteins will be associated with this traits in sheep. So we will focus on the up and down regulated proteins, especially some cytokines in the further research.

Finally, we try to improve the grammar and expression in the English writing of the manuscript before submit the revision. Special thanks to you for your good comments.

Yours sincerely

Bohui Yang

---

## [Decision Letter · Decision Letter 1]

18 Jan 2021

Quantitative proteomic analysis identified differentially expressed proteins with tail/rump fat deposition in Chinese thin- and fat-tailed lambs

PONE-D-20-24379R1

Dear Dr. Yang,

We’re pleased to inform you that your manuscript has been judged scientifically suitable for publication and will be formally accepted for publication once it meets all outstanding technical requirements.

Kind regards,

Livia D'Angelo

Academic Editor

PLOS ONE

Additional Editor Comments (optional):

Reviewers' comments:

Reviewer's Responses to Questions

**Comments to the Author**

1. If the authors have adequately addressed your comments raised in a previous round of review and you feel that this manuscript is now acceptable for publication, you may indicate that here to bypass the “Comments to the Author” section, enter your conflict of interest statement in the “Confidential to Editor” section, and submit your "Accept" recommendation.

Reviewer #1: All comments have been addressed

Reviewer #2: All comments have been addressed

2. Is the manuscript technically sound, and do the data support the conclusions?

Reviewer #1: Yes

Reviewer #2: Yes

3. Has the statistical analysis been performed appropriately and rigorously? 

Reviewer #1: Yes

Reviewer #2: Yes

4. Have the authors made all data underlying the findings in their manuscript fully available?

Reviewer #1: Yes

Reviewer #2: Yes

5. Is the manuscript presented in an intelligible fashion and written in standard English?

Reviewer #1: Yes

Reviewer #2: Yes

6. Review Comments to the Author

Reviewer #1: (No Response)

Reviewer #2: (No Response)

7. PLOS authors have the option to publish the peer review history of their article (what does this mean?). If published, this will include your full peer review and any attached files.

Reviewer #1: **Yes: **Adham Al-Sagheer

Reviewer #2: No

---

## [Editor Report · Acceptance letter]

22 Jan 2021

PONE-D-20-24379R1 

Quantitative proteomic analysis identified differentially expressed proteins with tail/rump fat deposition in Chinese thin- and fat-tailed lambs 

Dear Dr. Yang:

I'm pleased to inform you that your manuscript has been deemed suitable for publication in PLOS ONE. Congratulations! Your manuscript is now with our production department. 

Kind regards, 

on behalf of

Dr. Livia D'Angelo 

Academic Editor

PLOS ONE